# Extracellular Vesicles Derived from Kefir Grain *Lactobacillus* Ameliorate Intestinal Inflammation via Regulation of Proinflammatory Pathway and Tight Junction Integrity

**DOI:** 10.3390/biomedicines8110522

**Published:** 2020-11-20

**Authors:** Eun Ae Kang, Hye-In Choi, Seung Wook Hong, Seokwoo Kang, Hyeon-Young Jegal, Eun Wook Choi, Byung-Soon Park, Joo Sung Kim

**Affiliations:** 1Department of Internal Medicine and Institute of Gastroenterology, Yonsei University College of Medicine, Seoul 03722, Korea; cheerea@gmail.com; 2Prostemics Research Institute, Seoul 04778, Korea; hichoi@prostemics.com (H.-I.C.); ksw9213@prostemics.com (S.K.); hyjegal@prostemics.com (H.-Y.J.); ewchoi@prostemics.com (E.W.C.); 3Department of Internal Medicine and Liver Research Institute, Seoul National University College of Medicine, Seoul 03080, Korea; hswooki@gmail.com; 4Cellpark Clinic, Seoul 06029, Korea

**Keywords:** extracellular vesicle, *Lactobacillus*, experimental colitis, tight junction, NF-κB

## Abstract

The aim of this study was to demonstrate the anti-inflammatory effect of *Lactobacillus kefirgranum* PRCC-1301-derived extracellular vesicles (PRCC-1301 EVs) on intestinal inflammation and intestinal barrier function. Human intestinal epithelial cells (IECs) Caco-2 were treated with PRCC-1301 EVs and then stimulated with dextran sulfate sodium (DSS). Real-time RT-PCR revealed that PRCC-1301 EVs inhibited the expression of pro-inflammatory cytokines in Caco-2 cells. PRCC-1301 EVs enhanced intestinal barrier function by maintaining intestinal cell integrity and the tight junction. Loss of Zo-1, claudin-1, and occludin in Caco-2 cells and the colitis tissues was recovered after PRCC-1301 EVs treatment, as evidenced by immunofluorescence analysis. Acute murine colitis was induced using 4% DSS and chronic colitis was generated in piroxicam-treated IL-10^-/-^ mice. PRCC-1301 EVs attenuated body weight loss, colon shortening, and histological damage in acute and chronic colitis models in mice. Immunohistochemistry revealed that phosphorylated NF-κB p65 and IκBα were reduced in the colon tissue sections treated with PRCC-1301 EVs. Our results suggest that PRCC-1301 EVs may have an anti-inflammatory effect on colitis by inhibiting the NF-κB pathway and improving intestinal barrier function.

## 1. Introduction

Inflammatory bowel diseases (IBDs), including Crohn’s disease (CD) and ulcerative colitis (UC), are characterized by chronic recurrent intestinal inflammation. IBD is caused by multiple factors such as genetic predispositions, environmental factors, intestinal microbial changes, and excessive immune response [1,2,3]. Although the pathophysiology of IBD is incompletely understood, intestinal barrier function plays an important role in IBD pathogenesis [4,5,6]. Damage to intestinal epithelial cells compromises integrity and loosens tight junctions (TJs), leading to “leaky gut”. Defects in the intestinal epithelial barrier can facilitate direct interaction between pro-inflammatory antigenic components in the lumen and the intestinal epithelium [7]. Moreover, the nuclear factor kappa beta (NF-κB) signaling pathway is one of the major pathophysiologic mechanisms that can promote inflammatory cytokine production [8]. Current treatment options aim to suppress inflammation based on various mechanisms. However, the development of new therapeutic agents targeting various mechanisms associated with IBD is required to achieve unmet therapeutic goals.

Extracellular vesicle (EV) is a lipid bilayer-enclosed particle, which ranges from 20 to 500 nm in diameter [9,10]. EVs have several beneficial effects including a protective role in host intestinal epithelial cells [11]. An EV does not contain bacteria itself. Therefore, EVs cannot cause adverse effects such as bacteremia when administered as a therapeutic agent. Kefir grain is a fermented milk product containing various microbes such as *Lactobacillus* species and yeasts and has antimicrobial, immune regulatory, and anti-inflammatory effects [12,13,14].

Our previous study showed that EVs obtained from three kinds of kefir-derived *Lactobacillus* (*L. kefir*, *L. kefiranofaciens*, and *L. kefirgranum*) using ultracentrifugation can modulate inflammation responses via alleviating the production of inflammatory cytokines in tumor necrosis factor-α (TNF-α)-induced inflammation in Caco-2 cells and 2,4,6-trinitrobenzene sulfonic acid (TNBS)-induced colitis [15]. However, the mechanisms of the anti-inflammatory effect of EVs derived from kefir grains remain unclear. Furthermore, there are no reports of the protective activity of EVs derived from kefir grains on epithelial permeability or intestinal barrier integrity.

The aims of this study were to evaluate the anti-inflammatory effects of EVs derived from kefir grain *Lactobacillus*, PRCC-1301 EVs, on the experimental murine colitis model and cell lines, and its association with the NF-κB pathway and intestinal barrier function.

## 2. Materials and Methods

### 2.1. Preparation of PRCC-1301 EVs

*L. kefirgranum* PRCC-1301 was isolated from kefir grain; *L. keifrgraum* PRCC-1301 was cultured in glucose base medium for 17–24 h at 28–35 °C with gentle shaking. When the optical density of the culture at 600 nm reached 1.0, the bacteria were pelleted at 8000 rpm for 10 min, and the resulting supernatant was passed through a filter to remove any remaining cells. The isolation and concentration of extracellular vesicles were achieved with ultrafiltration [16]. The resulting solutions were dried. The dried products were then stored at room temperature until use.

### 2.2. Cell Culture

The human intestinal epithelial cells (IECs) Caco-2 and HCT116 were purchased from Korea Cell Line Bank (Seoul, Korea). The cells were cultured in 10% FBS (Gibco-BRL, Grand Island, NY)-supplemented DMEM (Hyclone, Logan, UT, USA), MEM/EBSS (Hyclone, Logan, UT, USA) with streptomycin (100 μg/mL)/penicillin (100 units/mL) (WelGene Inc., Gyeongsan-si, Korea) at 37 °C in a humidified chamber with 5% CO_2_.

### 2.3. Quantitative Real-Time Reverse Transcription PCR

Total RNA was isolated from Caco-2 cells, which were treated with PRCC-1301 EVs (100 μg/mL) for 6 h using RNeasy mini kit (QIAGEN, Valencia, CA, USA). Approximately 1 μg of total RNA was used for cDNA synthesis reaction using MaximeTM RT PreMix (iNtRON, Gyeonggi-do, Korea) according to the manufacturer’s instructions. For quantitative expression level of interleukin (IL)-2, IL-8, and TNF-α, real-time reverse transcription (RT) PCR was performed on StepOnePlus™ Real-Time PCR System (Applied Biosystems, Foster City, CA, USA) using SYBR Premix Ex Taq (TaKaRa Bio, Shiga, Japan) according to the manufacturer’s instructions. Primer pairs listed in Appendix A.

### 2.4. In Vitro Epithelial Monolayer Permeability Assay

Caco-2 and HCT116 cell monolayers were cultured on transwell chambers (Corning, New York, NY, USA) and stimulated on the basolateral side with 10 ng/mL TNF-α. PRCC-1301 EVs (0, 10, and 100 μg/mL) were added into the apical side and co-incubated for 48 h. Fluorescein isothiocyanate (FITC)-conjugated dextran were added to the apical compartment for 4 h. Fluorescence intensity was measured (excitation, 492 nm; emission, 525 nm) by 100 μL of media collected from the basal chamber.

### 2.5. Induction and Treatment of Colitis

Male wild-type (WT) C57BL/6 mice at 6 weeks of age (20–22 g) with specific pathogen-free (SPF) conditions were used for the DSS-induced acute colitis model (Young-Bio, Seongnam, Korea). Mice were divided into 4 groups; control group (*n* = 6), vehicle (*n* = 10), 0.03 mg/kg, and 3 mg/kg of PRCC-1301 EV-treated group (*n* = 10, respectively). PRCC-1301 EVs for the treated groups or PBS for the vehicle was administered by oral gavage from 2 day before administration of DSS to day 5, and 4% of dextran sulfate sodium (DSS) (Sigma-Aldrich, St. Louis, MO, USA) was administered with drinking water from day 0 to day 5. Mice were euthanized on day 5. Daily body weight changes, colon length, histopathological findings, and histologic scores through hematoxylin and eosin (H&E) staining were measured. Histologic scores were measured by combining the 4 categories; severity of inflammation, degree of injury, crypt damage, and proportion of tissue involvement [17,18].

The effect of PRCC-1301 EVs on the chronic colitis model was evaluated using IL-10^-/-^ C57BL/6 mice, 6–7 weeks of age and 19–22 g in weight. IL-10 is one of the anti-inflammatory cytokines and is known to develop chronic colitis over several months when IL-10 deficiency occurs. Piroxicam (200 ppm) was mixed with the feed and taken from day 0 to day 14 to aggravate colitis in IL-10^-/-^ mice [19]. Mice were divided into 4 groups; control group (*n* = 4), vehicle (*n* = 15), 0.03 mg/kg (*n* = 10), and 3 mg/kg of PRCC-1301 EV-treated group (*n* = 3). Each dose of PRCC-1301 EVs and PBS for vehicle was administered orally from day 14 to day 28. On day 28, mice were euthanized; weight changes, colon length and histologic characteristics were compared.

### 2.6. Immunofluorescence Analysis

Immunofluorescence analysis was performed on Caco-2 cells and paraffin-embedded colon tissue sections to evaluate the expression of ZO-1, claudin-1, and occludin (Thermo, Grand Island, NY, USA). After removal of paraffin, dehydration, and blocking, colon tissue was incubated with ZO-1, claudin-1, and occludin antibody (1:100) overnight at 4 °C. After washing with PBST three times, slides were incubated with FITC-labeled secondary antibody (Cell Signaling Technology, Beverly, MA, USA) for 1 h at room temperature. Slides were incubated with antibodies overnight at 4 °C and exposed to the FITC-conjugated secondary antibody at room temperature for 1 h. Slides were slightly counterstained with 4, 6-diamidino-2-phenylindole (DAPI) (Sigma-Aldrich, St. Louis, MO, USA), for nuclear counterstaining. EVOS^®^ FL Cell Imaging System (Thermo, Grand Island, NY, USA). FITC and DAPI images were taken from the same area.

Likewise, Caco-2 cells were treated with 2.5% DSS for 48 h with or without PRCC-1301 EV and fixed with PFA. After blocking, Caco-2 cells were incubated with the ZO-1 antibody (1:100) overnight at 4 °C After washing with PBST three times, cells were exposed to the FITC-conjugated secondary antibody (1:200) at room temperature for 2 h. Finally, the coverslips were stained with DAPI for 1 min and washed with PBST three times. Images were observed by EVOS^®^ FL Cell Imaging System (Thermo, Grand Island, NY, USA).

### 2.7. Immunohistochemical Analysis

Paraffin-embedded tissue sections were stained using AEC Kit (GBI Labs, Mukilteo, WA, USA) according to the manufacturer’s instructions. Briefly, after the removal of paraffin and dehydration, the tissue sections were washed in PBS, and the endogenous peroxidase activity was quenched using H_2_O_2_. Sections were incubated with primary antibodies against phospho-NF-κB p65 and phospho-IκBα (Cell Signaling Technology, Danvers, MA, USA) overnight at 4 °C. The sections were then washed in PBST and incubated with AEC chromogen. Slides were slightly counterstained with Mayer’s hematoxylin. Assessment of phospho-NF-κB p65 and phospho-IκBα was graded as following the intensity of immunoreactivity on a scale of 0 to 4+. Briefly, 0 means there were no positive cells (no); 1+ means there was < 10% positive cells (weak); 2+ means there were 10–30% positive cells (mild); 3+ means there were 31–60% positive cells (moderate); and 4+ indicated 61–100% positive cells (strong).

### 2.8. Statistical Analysis

All experiments were performed in triplicate for each condition and are expressed as the mean ± standard deviation. Statistical comparisons were analyzed using a one-way analysis of variance (ANOVA) and a Tukey’s multiple comparisons test. *p* values  <  0.5 were considered to indicate statistical significance. All statistical analyses were performed using GraphPad Prism (Version 5.0, GraphPad Software Inc., La Jolla, CA, USA).

### 2.9. Ethical Considerations

The Institutional Animal Care and Use Committees (IACUC) of Seoul National University (Seoul, Korea) approved this study on 2 February 2019 (IACUC No. SNU-181231-3). All relevant in vivo and in vitro studies were conducted according to the American Physiological Society (August 2010 revision).

## 3. Results

### 3.1. PRCC-1301 EVs Inhibit Pro-Inflammatory Cytokine Gene Expression in Caco-2 Cells

We investigated the effect of PRCC-1301 EVs on pro-inflammatory cytokine production in Caco-2 cells using real-time RT-PCR analysis. PRCC-1301 EVs showed significantly reduced IL-2, IL-8, and TNF-α gene expression in DSS-induced Caco-2 cells. (Figure 1A–C).

### 3.2. PRCC-1301 EVs Recover Increase in Intestinal Permeability and Disruption of Tight Junction Complexes

To determine the effect of PRCC-1301 EVs on epithelial permeability, we performed epithelial monolayer permeability assay in TNF-α-induced in Caco-2 and HCT116 cells. TNF-α-induced Caco-2 cells increase leakage of FITC-dextran compare with control. PRCC-1301 EVs significantly inhibited TNF-α-induced leakage of FITC-dextran in Caco-2 and HCT116 cells (Figure 2A,B).

Next, we investigated the protective effect of PRCC-1301 EVs on the epithelial barrier, using the Caco-2 cell line. To form the intestinal barrier model, Caco-2 cells were cultured to 100% confluency to form a monolayer. As shown in Figure 2C,D, the TJ structure was disrupted and ZO-1 expression was decreased in DSS induction; however, PRCC-1301 EVs protected TJs from DSS-induced damage. We also investigated the protective effect of PRCC-1301 EVs against DSS-induced disruption of TJs; we examined the expression of TJ proteins including ZO-1, claudin-1, and occludin using immunofluorescence assay in DSS-induced colitis mouse colon tissue. The DSS-treated group showed disruption of the intestinal intercellular structure and considerable loss of ZO-1, claudin-1, and occludin. Interestingly, PRCC-1301 EVs treatment significantly prevented the loss of ZO-1, claudin-1, and occludin in TJ proteins of colonic epithelial cells (Figure 2E,F). Overall, these results suggest that treatment with PRCC-1301 EVs increased the expression of TJ proteins, potentially enhancing the intestinal barrier integrity and decreased epithelial permeability.

### 3.3. PRCC-1301 EVs Prevent DSS-Induced Acute Colitis in Mice

The preventive effect of PRCC-1301 EVs on acute colitis was evaluated using a DSS-induced colitis model in WT mice. DSS can cause inflammation of the colon mucosa, similar to UC. PRCC-1301 EV-treated group showed less weight loss (Figure 3A) and recovered colon length compared to the vehicle (Figure 3B). Based on H&E staining and histologic scores of colon inflammation, PRCC-1301 EVs attenuated epithelial damage, crypt loss, and leukocyte infiltration to the lamina propria and submucosa compared to the vehicle (Figure 3C).

### 3.4. PRCC-1301 EVs Attenuate Chronic Colitis in IL-10^-/-^ Mice

Chronic colitis in IL-10^-/-^ mice was one of the colitis models similar to CD. PRCC-1301 EVs were administered to the piroxicam-induced chronic murine colitis model for 14 days using IL-10^-/-^ mice. There was no significant difference in body weight change over 28 days between the vehicle- and PRCC-1301 EV-treated groups (Figure 3D). However, in the PRCC-1301 EV-treated group, shortening of the colon length was reduced, and histopathologic findings and histologic score demonstrate the therapeutic effect of PRCC-1301 EVs compared to the vehicle (Figure 3E,F).

These results indicate that the PRCC-1301 alleviate DSS-induced colitis and chronic colitis in IL-10^-/-^ models.

### 3.5. PRCC-1301 EVs Inhibit the NF-κB Pathway in the Colitis Mucosa

To investigate whether PRCC-1301 EVs inhibit NF-κB activation, we performed immunohistochemical staining of the phosphorylated NF-κB p65 in intestinal epithelial cells of DSS-induced colitis mice. The number of phospho-NF-κB p65 positive cells was reduced in the PRCC-1301 EV-treated group than in the vehicle-treated group in the distal colon as well as the proximal colon (Figure 4A,B). We also examined the phospho-IκBα expression in colon tissues in DSS-induced colitis, and the results showed that phospho-IκBα expression was decreased in the PRCC-1301 EV-treated group than in the vehicle-treated group (Figure 4C,D).

Taken together, these results indicated that the NF-κB signaling pathway might be inhibited by PRCC-1301 EVs.

## 4. Discussion

Certain probiotics exhibit immunomodulatory activity and directly affect the expression of cytokines and intestinal permeability by reducing inflammation and protecting intestinal barrier integrity [20]. However, even though many studies have reported an effect of probiotics on IBD, there has been very little research reported on the effectiveness of *Lactobacillus*-derived EVs.

As previously mentioned, we report for the first time that kefir-derived *Lactobacillus* EVs suppressed the production of inflammatory cytokines in Caco-2 cells and the TNBS-induced colitis model. In this study, to improve the bioavailability in IBD treatment, we isolated and identified a new strain of *Lactobacillus kefirgranum* PRCC-1301 and purified EVs using the filtration method for eliminating toxicity and increasing purity. Additionally, we investigated whether PRCC-1301 EVs alleviate DSS-induced colitis and chronic colitis in IL-10^-/-^ models and protect the epithelial barrier in Caco-2 cells.

EVs isolated from *Lactobacillus* spp. exert a suppressive effect on various pro-inflammatory cytokines. *L. paracasei*-derived EVs reduced the expression of the LPS-induced pro-inflammatory cytokines IL-1α, IL-1β, IL-2, and TNFα [21]. In addition, *L. plantarum*-derived EVs reduce inflammatory responses in S. *aureus*-induced atopic dermatitis mice by suppressing IL-6 and IL-4 [22]. In addition, *L. plantarum* and *L. rhamnosus*-derived EVs inhibit the production of TNF-α and IL-6 in LPS-induced RAW 264.7 mouse macrophages [23]. These findings support the anti-inflammatory capability of various *Lactobacillus*-derived EVs.

It is well known that inflammatory conditions of the intestinal mucosa lead to compromised barrier function. The balance of pro- and anti-inflammatory cytokines in the colonic mucosa is essential for intestinal homeostasis [24,25]. TNF-α or IFN-γ can increase the epithelial permeability by regulating TJs and promoting apoptosis [26]. IL-17A, a major Th17 pro-inflammatory cytokine, which plays an important role in the crosstalk between innate and adaptive immunity, promotes production of IL-8 in epithelial cells and stimulates recruitment of neutrophils and additional Th17 cells to the site of inflammation [27,28]. The results of this study suggested that PRCC-1301 EVs inhibited pro-inflammatory cytokines in colon epithelial cells and that they would be effective against IBD.

The major effect of the IBD pathogenesis is the impairment of desmosomes, particularly the TJs that maintain intercellular adhesion and regulate paracellular transport [29]. In previous studies, the treatment of epithelial monolayers with *Lactobacillus* spp. (*L. rhamnosus* GG, *L. plantarum*, and *L. salivarius*) enhanced protein expression of the TJ proteins (ZO-1, ZO-2, and cingulin) and other TJ-related proteins (claudin-1, claudin-3, and JAM-1) [30,31,32]. Hsieh et al. demonstrated that some *Bifidobacterium* and *Lactobacillus* species protect the epithelial TJ barrier against TNF-α-induced injury and promote the restoration of TNF-α-induced loss of epithelial barrier integrity. In addition, the heat-killed *B. bifidum* WU12 lost the TJ repair capacity completely [33]. This suggests that the main inducing factor of this beneficial effect might be the metabolites of bacteria, such as EVs, not the cell wall components. Interestingly, microbiota-derived EVs from both probiotic Escherichia coli Nissle 1917 (EcN) and ECOR63 promote the up-regulation of ZO-1 and claudin-14 and down-regulation of claudin-2, thus reinforcing the epithelial barrier, while the specific mechanism has not been illustrated [34]. PRCC-1301 EVs reduced epithelial barrier permeability in TNF-α-induced Caco-2 and HCT116 cells, by suppression of FITC-dextran uptake. Intestinal permeability increases with age in mice due to age-related microbial dysbiosis, inflammation, and barrier function disruption. We also confirmed that PRCC-1301 EVs reduced intestinal permeability in a 19-month-old mouse model.

In several studies, IFN-γ and TNF-α have been found to induce the internalization of the apical-junctional complex by increasing macropinocytosis and down-regulating TJ proteins [35,36,37]. IFN-γ and TNF-α also reorganize numerous TJ proteins, which increase intestinal permeability [38] and contribute to IBD. The DSS-induced Caco-2 cell model can be used to study the mechanism of DSS contacting the intestinal epithelium, similar to the early stages of colitis [39]. PRCC-1301 EVs prevented DSS-induced loss of ZO-1 in Caco-2 monolayer. The in vivo experiment in this study showed that PRCC-1301 EVs recovered the expression and distribution of TJ proteins, ZO-1, claudin-1, and occludin in DSS-induced colitis. Our data showed that PRCC-1301 EVs might have a protective effect on barrier integrity by maintaining the expression of TJ proteins, thereby reducing the severity of colitis.

NF-κB signaling pathway is the major pathway that regulates the expression of pro-inflammatory cytokines that are involved in the development of IBD [40]. NF-κB p65 is located in the cytoplasm and binds to IκBα as an inactive complex [41,42]. The phosphorylation and subsequent degradation of IκBα result in separation of the complex, and then NF-κB is activated. The activated NF-κB promotes the expression of inflammatory cytokines. [43]. Our data showed that PRCC-1301 EVs downregulated NF-κB signaling pathways in colon tissues by suppressing phosphorylation of IκBα in DSS-induced and IL-10^-/-^ colitis. PRCC-1301 EVs might have anti-inflammatory effects through regulation of phosphorylation/degradation of IκBα in NF-κB signaling pathway.

Above all, our study may broaden our current understanding of PRCC-1301 EVs and their beneficial effects using the cell line and colitis mouse model. We could provide a hypothesis that PRCC-1301 could prevent the disruption of TJ structure and TJ-associated proteins and inhibit the proinflammatory pathway, thus preventing the increase in the intestinal permeability and decrease the colonic inflammation.

## 5. Conclusions

In conclusion, PRCC-1301 EVs may have an anti-inflammatory effect on colitis, improving intestinal barrier function. PRCC-1031 EVs can be considered as a new therapeutic candidate in IBD.

## Figures and Tables

**Figure 1 biomedicines-08-00522-f001:**
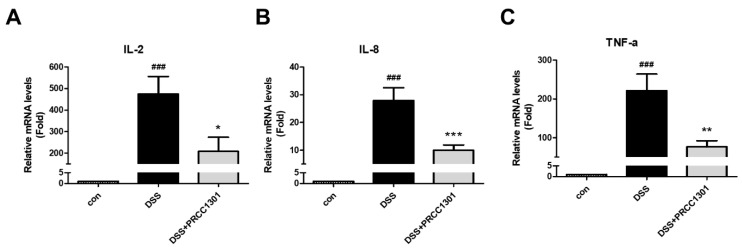
PRCC-1301 extracellular vesicles (EVs) inhibited pro-inflammatory cytokine gene expression in Caco-2 cells. (**A**) mRNA expression level of (**A**) IL-2, (**B**) IL-8, and (**C**) TNF-α in Caco-2 cells. The results are shown as mean ± SEM. ^###^
*p* < 0.001 compared with control, * *p* < 0.05 and ** *p* < 0.01 and *** *p* < 0.001 compared with DSS group.

**Figure 2 biomedicines-08-00522-f002:**
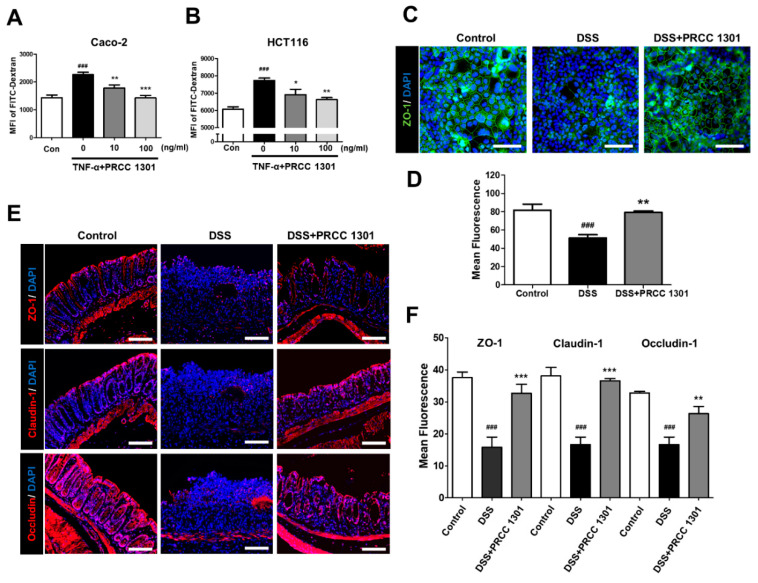
PRCC-1301 EVs recovered increase in intestinal permeability and disruption of tight junction complexes. In vitro permeability assay was performed in (**A**) Caco-2 and (**B**) HCT116 cells treated with TNF-α in the absence or presence of PRCC-1301 EVs. The results are shown as mean ± SEM. ^###^
*p* < 0.001 compared with control, * *p* < 0.05, ** *p* < 0.01, and *** *p* < 0.001 compared with TNF-α-treated group. (**C**) Expression of ZO-1 in Caco-2 cell monolayers incubated with 2.5% DSS in the absence or presence of PRCC-1301 EVs. (**D**) Quantification for ZO-1 fluorescence. The results are shown as mean ± SEM. ^###^
*p* < 0.001 compared with control, ** *p* < 0.01 compared with DSS group. (**E**) Expression of ZO-1, claudin-1, and occludin was examined in control, DSS, and PRCC-1301 EV-treated mouse colon tissue. (**F**) Quantification for ZO-1, claudin-1, and occluding fluorescence, respectively. Nuclei were counterstained with 4, 6-diamidino-2-phenylindole (DAPI). The results are shown as mean ± SEM. ^###^
*p* < 0.001 compared with control, ** *p* < 0.01 and *** *p* < 0.001 compared with DSS group. Scale bar = 100 µm.

**Figure 3 biomedicines-08-00522-f003:**
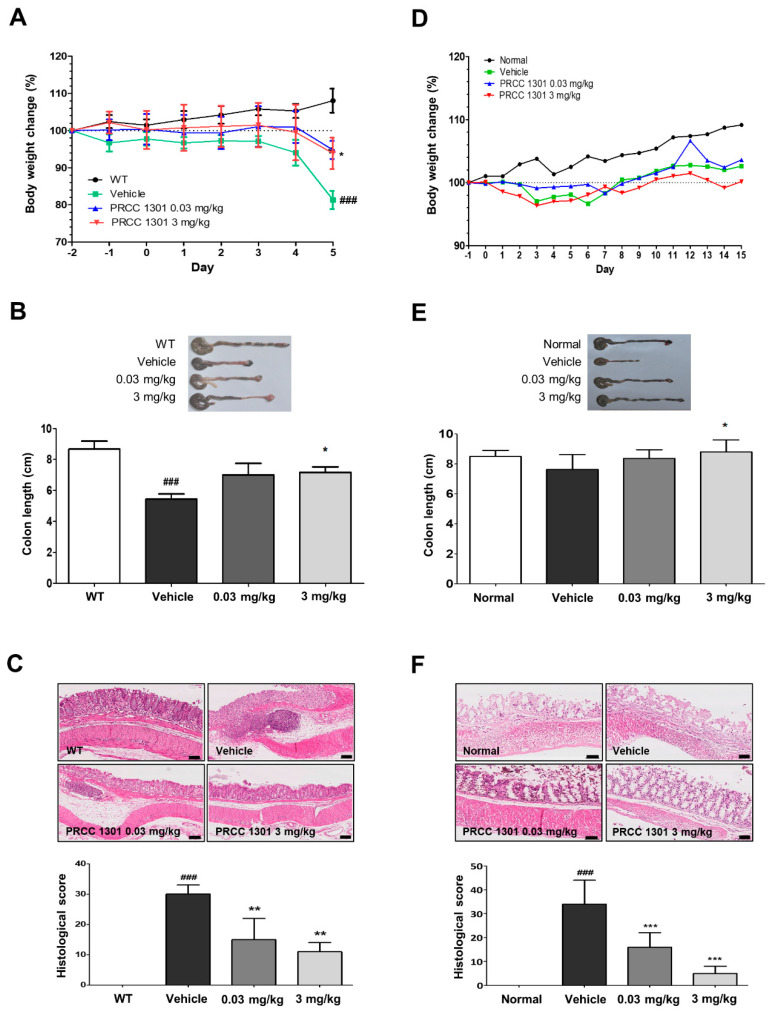
PRCC-1301 EVs prevented DSS-induced acute colitis and attenuated chronic colitis in IL10^-/-^. (**A**,**D**) Body weight, (**B**,**E**) colon length, and (**C**,**F**) histological evaluation in DSS‑induced and IL10^-/-^ colitis, respectively. The results are shown as mean ± SEM. ^###^
*p* < 0.001 compared with wild-type (WT), * *p* < 0.05, ** *p* < 0.01, and *** *p* < 0.001 compared with vehicle group. Scale bar = 100 µm.

**Figure 4 biomedicines-08-00522-f004:**
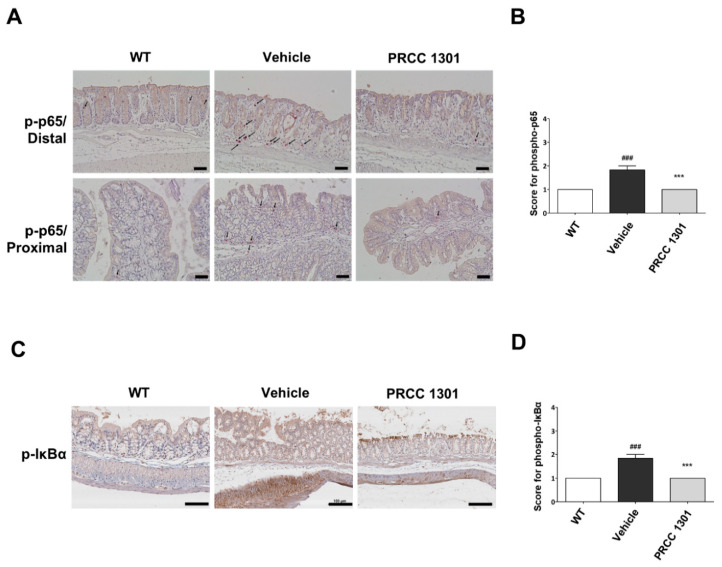
PRCC-1301 EVs suppressed NF-κB activation in DSS-induced acute colitis. Representative images of Immunohistochemistry (IHC) staining of phospho-NF-κB p65 and phospho-IκBα in 3 mg/kg of PRCC-1301 EV-treated mice. (**A**,**B**) The expression of phospho-NF-κB p65 in DSS-induced colitis mice. The arrow indicates phospho-NF-κB p65 positive cells. The expression of phospho-IκBα in (**C**,**D**) DSS-induced colitis. The results are shown as mean ± SEM. ^###^
*p* < 0.001 compared with WT, *** *p* < 0.001 compared with vehicle group. Scale bar = 100 µm.

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
