# Peer review of "Extracellular Vesicles Derived from Kefir Grain Lactobacillus Ameliorate Intestinal Inflammation via Regulation of Proinflammatory Pathway and Tight Junction Integrity"

_biomedicines, 2020, doi:10.3390/biomedicines8110522_

Round 1
Reviewer 1 Report
The authors are presenting an interesting study on the effect of extracellular vesicles from Lactobacillus kefirgranum on intestinal barrier in inflammatory bowel disease (IBD). This approach can be used to develop a new probiotic therapy for IBD.
However, there are several concerns that need to be addressed before the publication, as listed below:
- Cytokine tested are not uniformed in Caco-2 and RAW cells. It is questionable whether the authors couldn't reach a similar results for cytokines in both cell types.
- The statistics in the results from RAW cells are questionable. According to the graph and standard deviation, there should not be any statistical significance.
- Figure 2C; Are the magnifications of the images the same? The control seems to have smaller cell area and weaker signal overall.
- Fig 3E: the image showing significantly shorter intestine for the mice treated with vehicle doesn’t fit with the represented graph .
- Fig 4: The staining of phospho-nfkb and phospho-Ikba is not definite, since it is very hard to differentiate the nucleus and the positive staining. Another way for quantification would be needed (such as western blot)
- More experiment details need to be added to materials and methods section. For example the treatment duration of the cells with the compound (4.3) and when the time when fluorescein permeability was measured, and whether the experiment was static or kinetic (4.4). More details on antibodies used and fluorescence microscopy system will also need to be added (4.6).
Author Response
Reviewer #1:
The authors are presenting an interesting study on the effect of extracellular vesicles from Lactobacillus kefirgranum on intestinal barrier in inflammatory bowel disease (IBD). This approach can be used to develop a new probiotic therapy for IBD.
However, there are several concerns that need to be addressed before the publication, as listed below:
- Cytokine tested are not uniformed in Caco-2 and RAW cells. It is questionable whether the authors couldn't reach a similar result for cytokines in both cell types.
Reply:
Thank you for your comments. Caco-2 cells are human intestinal epithelial cells and RAW 264.7 are mouse macrophages. We induced inflammation with different substances depending on the cell type and identified representative cytokines for each inducer. The same cytokine was not tested in both cell types because it was reported that the IL-8 gene was deleted in mice.
- The statistics in the results from RAW cells are questionable. According to the graph and standard deviation, there should not be any statistical significance.
Reply:
Thank you for your precious comments. In figure 1, we used a paired two-tailed Student's t-test, however, all other data were analyzed using the one-way analysis of variance (ANOVA) and Tukey's multiple comparisons test. As pointed out by the reviewer, the experimental results of RAW 264.7 cells were judged to have no statistical significance, so we decided to delete the experimental results of RAW264.7 cells (Figure 1D, 1E). In addition, statistics from all experiments were consolidated into the one-way analysis of variance (ANOVA) and Tukey’s multiple comparisons test.
- Figure 2C; Are the magnifications of the images the same? The control seems to have smaller cell area and weaker signal overall.
Reply:
Thank you for your critical comments. All images have the same magnification. To form the intestinal barrier model, Caco-2 cells were cultured for 7 days or longer to form a monolayer. In this process, well-differentiated cells form tight junctions between cells, and differentiated cells have an enlarged nucleus. As for the control, the monolayer was well-formed and the nucleus of the cell became enlarged. The tight junction structure was disrupted in DSS induction, and the cell is also affected and the size of the nucleus changes. However, PRCC-1301 EVs protected TJs from DSS-induced damage. In the same experimental model, we add reference data in which the size of the cell nucleus varies according to DSS induction.
(Chen, L. et al., Activating AMPK to Restore Tight Junction Assembly in Intestinal Epithelium and to Attenuate Experimental Colitis by Metformin, Front Pharmacol. 2018 Jul 16;9:761.)
- Fig 3E: the image showing significantly shorter intestine for the mice treated with vehicle doesn’t fit with the represented graph.
Reply:
Thank you for your comment. Images of colon specimens were chosen as the most representative features in each group. However, the graph was created using the average colon length between all mice in each group. Therefore, the picture and graph of the colon look different. As can be seen from the graph, the shortening of the colon length showed a statistical difference between groups when comparing the mean and standard deviation.
- Fig 4: The staining of phospho-nfkb and phospho-Ikba is not definite, since it is very hard to differentiate the nucleus and the positive staining. Another way for quantification would be needed (such as western blot)
Reply:
I agree with the reviewer's opinion. It is good to quantify NF-kB through western blot, but the experiment was difficult to be performed due to the limited re-submission time. The relevance of NF-kB was supplemented in the discussion. Also, the title has been changed from “Extracellular vesicles derived from kefir grain Lactobacillus ameliorate intestinal inflammation via regulation of NF-kB pathway and tight junction integrity” to “Extracellular vesicles derived from kefir grain Lactobacillus ameliorate intestinal inflammation via regulation of proinflammatory pathway and tight junction integrity”.
- More experiment details need to be added to the materials and methods section. For example, the treatment duration of the cells with the compound (4.3) and when the time when fluorescein permeability was measured, and whether the experiment was static or kinetic (4.4). More details on antibodies used and fluorescence microscopy system will also need to be added (4.6).
Reply:
Thank you for your comments. We have added detailed experimental methods to materials and methods.

Reviewer 2 Report
In this paper, Kang et al. analyze the anti-inflammatory effect of kefir extracellular vesicles in the context of intestinal inflammation.
The anti-inflammatory effects of kefir have been widely analyzed in different scenarios, and the authors of this paper have a previous work in which they have already described the effect of kefir extracellular vesicles in the inhibition of intestinal inflammation. In this paper they go a little further in the mechanistic characterization of kefir anti-inflammatory effects and evaluate the involvement of the NFkB pathway.
Specific comments:
- In figure 2 (panels C and D) some quantification from different fields/experimental replicates are needed.
- In figure 3 vehicle is compared to control, and treatments (0,03mg/kg and 3mg/kg) are compared to vehicle. Treatments should be also compared to vehicle in order to know if there are differences between these conditions. Same in figure 4.
- Concluding that EVs regulate NFkB with the data they show is too much of a conclusion. They will need at least a western blot and I suggest they evaluate the presence of NFkB in cytoplasmic and nuclear compartments.
Author Response
Reviewer #2:
In this paper, Kang et al. analyze the anti-inflammatory effect of kefir extracellular vesicles in the context of intestinal inflammation.
The anti-inflammatory effects of kefir have been widely analyzed in different scenarios, and the authors of this paper have a previous work in which they have already described the effect of kefir extracellular vesicles in the inhibition of intestinal inflammation. In this paper they go a little further in the mechanistic characterization of kefir anti-inflammatory effects and evaluate the involvement of the NFkB pathway.
Specific comments:
- In figure 2 (panels C and D) some quantification from different fields/experimental replicates are needed.
Reply:
Thank you for your critical comments. We added data from different experimental replicates for Figures 2D and 2F.
- In figure 3 vehicle is compared to control, and treatments (0,03mg/kg and 3mg/kg) are compared to vehicle. Treatments should be also compared to vehicle in order to know if there are differences between these conditions. Same in figure 4.
Reply:
Thank you for your comment. A comparison of DSS-treated vehicle and control showed that the induction of inflammation was successful in this experiment. In addition, the vehicle and each treatment group were compared at different doses to evaluate a therapeutic effect of the extracellular vesicles. All comparison results between groups analyzed by ANOVA and Tukey’s multiple comparison tests are presented below.
|
Tukey's Multiple Comparison Test |
Mean Diff. |
q |
Significant? P < 0.05? |
Summary |
95% CI of diff |
|
Control vs Vehicle |
-29.25 |
10.06 |
Yes |
*** |
-41.83 to -16.67 |
|
Control vs PSI-401 0.03mg/kg |
-14.75 |
5.072 |
Yes |
* |
-27.33 to -2.167 |
|
Control vs PSI-401 3mg/kg |
-11.00 |
3.783 |
No |
ns |
-23.58 to 1.583 |
|
Vehicle vs PSI-401 0.03mg/kg |
14.50 |
6.107 |
Yes |
** |
4.226 to 24.77 |
|
Vehicle vs PSI-401 3mg/kg |
18.25 |
7.686 |
Yes |
** |
7.976 to 28.52 |
|
PSI-401 0.03mg/kg vs PSI-401 3mg/kg |
3.750 |
1.579 |
No |
ns |
-6.524 to 14.02 |
- Concluding that EVs regulate NFkB with the data they show is too much of a conclusion. They will need at least a western blot and I suggest they evaluate the presence of NFkB in cytoplasmic and nuclear compartments.
Reply:
I agree with the reviewer's opinion. It is good to quantify NF-kB through western blot, but the experiment was difficult to be performed due to the limited re-submission time. The relevance of NF-kB was supplemented in the discussion. According to our previous study related to kefir-derived exosome, the levels of phospho-p65 were decreased in the treated group (MK Seo et al., Therapeutic effects of kefir grain Lactobacillus-derived extracellular vesicles in mice with 2,4,6-trinitrobenzene sulfonic acid-induced inflammatory bowel disease, J Dairy Sci. 2018 Oct;101(10):8662-8671.).
Also, the title has been changed from “Extracellular vesicles derived from kefir grain Lactobacillus ameliorate intestinal inflammation via regulation of NF-kB pathway and tight junction integrity” to “Extracellular vesicles derived from kefir grain Lactobacillus ameliorate intestinal inflammation via regulation of proinflammatory pathway and tight junction integrity”.
